# Increasing Population Status of Broad-Snouted Caiman (*Caiman latirostris*) Based on Sustainable Use Strategies in a Managed Protected Area in Santa Fe, Argentina

**DOI:** 10.3390/ani14091288

**Published:** 2024-04-25

**Authors:** Pablo Ariel Siroski, Hernán Ciocan, Samuel Hilevski, Alejandro Larriera

**Affiliations:** 1Laboratorio de Ecología Molecular Aplicada, Instituto de Ciencias Veterinarias del Litoral, Universidad Nacional del Litoral/Consejo Nacional de Investigaciones Científicas y Técnicas, Esperanza 3080, Santa Fe, Argentina; samuel.hilevski@icivet.unl.edu.ar; 2Ministerio de Ambiente y Cambio Climático, Gobierno de Santa Fe, Santa Fe 3000, Santa Fe, Argentina; hernanciocan@gmail.com; 3Laboratorio de Zoología Aplicada, Universidad Nacional del Litoral (FHUC/UNL/MAyCC), Santa Fe 3000, Santa Fe, Argentina; alelarriera@gmail.com

**Keywords:** ranching, local communities, sustainable use, consumptive use, management

## Abstract

**Simple Summary:**

Broad-snouted caimans are found in North Argentina. In the past, they were heavily hunted, leading to a decline in their population. To avoid diminishing populations, a sustainable use program under a ranching system was implemented, and broad-snouted caiman populations have recovered significantly. Local communities play a key role as egg harvesters under the concept of incentives for conservation. The current management program in Argentina intends to balance the growth of caiman populations, ensure benefits for local people, and protect valuable wetlands. Each year, the dynamics of the broad-snouted caiman population are monitored through night surveys and the identification of wild nesting in specific areas. These monitoring efforts show that the sustainable use program ensures practices that are sustainable and have a non-detrimental impact on wild populations. On the contrary, the encouragement of local people to preserve caimans and their nesting habitats has generated very positive results, including increased caiman populations and improvements in social and cultural aspects, as well as providing important income to local economies. There are many precedents regarding the advantages of enacting sustainable use programs, but this knowledge is not yet adequately communicated. Even crocodilians are an example of this, and we already have numerous studies about the pillars that programs should follow, regardless of the species and management mechanism implemented. Now the great challenge is to maintain the achievements reached and control threats such as habitat destruction, pollution, and activist groups from reversing them.

**Abstract:**

People and crocodilians have engaged in interactions since the earliest human settlements. After many years of escalating non-regulated use, coupled with emerging threats such as habitat modification, environmental pollution, and the exponential growth of human populations, natural crocodilian populations have been pushed to the brink of extinction. To prevent this, various initiatives have implemented strategies to prevent local extinction. Reinforcing wild populations through reintroductions and head-starting programs has been considered the safest approach to recovering crocodilian populations. Subsequently, the concept of sustainability emerged. In the case of many historically exploited crocodilian species, it became evident that rational utilization need not adversely affect population status; on the contrary, there were clear signs of recovery when local communities were involved. In 1990, the broad-snouted caiman (*Caiman latirostris*) was in a critical state due to decades of poaching and the aforementioned threats. In response, the “Monitoring and restocking program of the broad-snouted caiman for management purposes”, commonly known as Proyecto Yacaré, was established to study population abundance and assess the biological, ecological, and social response to this management methodology. As a result of the eggs’s harvest, a proportion equivalent to or greater than what would have survived in natural conditions was released into the original habitat where the eggs were collected. The surplus was allocated for leather and meat production with a focus on economic and environmentally sustainable practices, without affecting biodiversity or threats to the managed species. The in situ work carried out by this program has been crucial for the recovery of broad-snouted caiman populations under the “incentives for conservation” system, whereby local communities receive benefits for nest identification and egg harvesting. Over time, conservation incentives have become significant contributors to regional economies. This study illustrates how *C. latirostris* populations increased due to the implementation of egg harvesting by local communities in a natural reserve for management purposes. This population increase was detected through continuous night monitoring and an analysis of the number of nests in the area. Currently, *C. latirostris* populations have transitioned from being among the most threatened to becoming one of the most abundant throughout their distribution area. Based on the analyzed information, we may affirm that the commercial value of these species is one of the most relevant aspects contributing to the sustainability of these programs, primarily due to the change in perception generated among local communities. Therefore, identifying and controlling factors affecting these programs are fundamental for the conservation of these species.

## 1. Introduction

There are 27 extant crocodilian species and subspecies of crocodilians distributed across 99 countries, ranging from one or two species to six species per country [1]. They are present in a wide variety of freshwater habitats, and they are exposed to the most diverse contexts that we can imagine, even with the same species in different countries.

Broad-snouted caiman (*Caiman latirostris*) and yacare caiman (*Caiman yacare*) are the two species of crocodilians that inhabit Argentina, and they can be easily morphologically differentiated based on their cranial and post-occipital scute structures [2]. Both species are sympatric in a large part of their distribution in Argentina, but due to their more southern and western distribution range, the broad-snouted caiman occurs allopatrically in a larger area [3]. *Caiman latirostris* prefers habitats that are densely vegetated and bask on floating vegetation or reservoirs, while *C. yacare* generally occurs in environments free of vegetation and tends to bask on beaches or coasts [2,4].

In the past, the exploitation of various crocodilian species represented a relevant economic activity in some regions of the world, and due to the excessive extraction to which crocodilians were subjected, many of them were seriously endangered [5]. There was a differential use among the caiman species living in Argentina since due to the greater ossification of the osteoderms of the yacare caiman, the most demanded was the broad-snouted caiman [6]. In 1990, the broad-snouted caiman was in a critical conservation situation as a result of non-regulated poaching in the last decades and was under increasing threat from the sources mentioned above [7]. Contrarily, but less significant for population impact, some wild adult caiman were regularly killed by the local inhabitants, sometimes to sell the skin on the illegal market but also out of fear for the welfare of small animals and even for meat consumption. Regarding its conservation status, it was classified as Low risk/Least concern (Lr/Lc) by the International Union for Conservation of Nature (IUCN) and listed in Appendix I of the Convention on International Trade in Endangered Species of Wild Fauna and Flora (CITES) from 1977 to protect it.

When most crocodilian species were considered threatened, and their populations were declining in both abundance and distribution based on non-regulated extraction, protection and coordinated efforts to prohibit illegal trade yielded results and helped populations recover quickly [8]. If habitats are in good condition, crocodilians respond positively to those actions, wild populations recover in abundance and distribution, and the average size of individuals increases in the population [9]. At present, there are emerging alternatives that integrate traditional crocodilian practices such as closed-cycle captive breeding and ranching, which engage local communities and promote habitat conservation through tangible incentives.

There was no record of commercial harvesting of caimans in Argentina in the late 1980s and early 1990s. However, the decline in caiman populations is not solely attributed to poaching; it is also exacerbated by ongoing habitat loss. Environmental alterations like deforestation, wetland drainage, and, more recently, intensive agriculture are identified as significant contributors to the issue [10].

Some approaches, such as ranching eggs combined with head-started restocking, are seen as the most secure methods of restoring crocodilian populations.

There are species with commercial value and those without in terms of trade. Species facing decline and potential extinction exist in both categories, but paradoxically, fewer species with commercial value through trade and under sustainable use programs are at risk [6]. Subsequently, the concept of sustainability emerged, revealing that rational and regulated utilization of historically exploited crocodilian species does not necessarily affect population status. On the contrary, clear signs of recovery are evident when local communities are involved in an economic context.

The IUCN recognizes the sustainable use of wildlife resources as a key incentive for species conservation and habitat protection.

Several models have emerged to encourage the use of wild ecosystems through trade, providing an incentive for conservation efforts. Many management programs operate on the premise that the economic value derived from the use of wildlife, whether consumptive or non-consumptive, can incentivize conservation efforts and habitat preservation. In this context, a ranching program for broad-snouted caimans, involving egg harvesting for subsequent management in captivity, was implemented in 1990 [7]. The program, commonly known as “Proyecto Yacaré”, was officially titled “Monitoring and Head Starting Program of the Broad-Snouted Caiman (*Caiman latirostris*) for Management Purposes”. Their objectives included determining the abundance of wild populations and evaluating the biological, ecological, and social responses to this wildlife management approach. The program involved the collection of eggs from the wild, which were then transferred to an artificial incubator. After hatching, the animals were raised under controlled temperatures and feeding conditions to promote rapid growth. When they reached around nine to ten months old, all of them were released back into the same locations where the eggs were collected.

At the 10th CITES Conference of the Parties in Harare, Zimbabwe, in 1997, Argentinean populations of *Caiman latirostris* were transferred to Appendix II, allowing international trade of products under ranching. Sustainable use operations were initially implemented in the Santa Fe province. The decision to downlist the species in CITES was primarily based on scientific data generated by the ranching program developed in Santa Fe province. Argentina and Brazil are the only countries within the species’ entire range listed under CITES Appendix II, with the exception of the Brazilian population, which is under a zero quota.

Since its beginning in 1990, the ranching program in Santa Fe province has released approximately 30,000 head-started *C. latirostris* yearlings into the wild. The restoration of the wild population has been observed with remarkable increases of up to 1500% in certain areas. Additionally, it has been documented that 50% of the breeding females in the operational zones are individuals previously reintroduced by the program [11].

The egg harvest is conducted by local individuals known as “gauchos”, following the technique proposed by Larriera [9]. In some cases, the gauchos locate the nests but are unable to collect the eggs for various reasons, and the gauchos’ family also contribute to egg collection at different times [12]. Local communities are involved in ranching because it brings in additional income for them [8,13].

Sustainable use in crocodilians has been consistently achieved, leading to a notable recovery in the populations of some of the most commercially valuable species, which are no longer facing extinction threats. It is essential to assess caiman populations and evaluate the effectiveness of existing sustainable use programs by monitoring them over time to demonstrate their success. Key parameters for evaluation include population abundance and structure, and if possible, comparing the conservation status of the population over time in similar study areas [14]. Accurately estimating these parameters is crucial for making informed decisions regarding the implementation of management programs and sustainable use [15,16]. However, conducting these estimates is a challenge due to numerous variables that can impact the accuracy of the methods used [17].

To measure the effects of sustainable use implementation on *C. latirostris* wild populations, an assessment of egg harvesting records and long-term surveys were conducted in a natural reserve in Santa Fe province as a reference place for ranching activities.

## 2. Materials and Methods

Study Area: Managed nature reserves are areas that have been designated for the preservation of specific sites or habitats that are crucial for maintaining populations of species of conservation importance or for sustainable use by local communities. These reserves allow for some level of environmental management to create optimal living conditions for the targeted species or communities. This site is considered the flagship location of the Proyecto Yacaré and serves as a central reference point for researchers specializing in this subject.

However, it is worth noting that identifying nests can be challenging in certain habitats. While nests are typically easier to detect in savannas and forests, they are less commonly found in habitats with floating vegetation [18]. The populations of broad-snouted caiman in the El Fisco Reserve are considered relatively stable (Figure 1, Figure 2 and Figure 3), and the channel and the lagoon both lack areas with floating vegetation suitable for nesting. Depending on the water level of the lagoon, the animals either move towards the channel or remain in the lagoon.

Population Surveys: Spotlight surveys were conducted from 1996 to 2018 throughout the channel and the lagoon of El Fisco Natural Reserve along a maximum of 12.3 km of the survey route. It is important to highlight that the environmental conditions, time of year, and sampling site were consistent across all monitoring efforts. Since the beginning of the Proyecto Yacaré activities in the El Fisco Reserve, assessments of the population status of *C. latirostris* have been conducted using the nocturnal count method. Boat-based sampling was undertaken at night, counting the total number of animals by detecting the reflection of their eyes illuminated with powerful lights. Additionally, efforts were made to approach the animals as closely as possible to classify them by size class.

The surveying procedures involved navigating the lagoon along its navigable section (water mirror) on some days, alternating with days when the canal was surveyed. The counts entailed the person quantifying the number of animals and striving to position themselves as closely as possible to identify them by size class, adhering to the criteria established by Ross and Godshalk [19]. According to their total length classification, animals were categorized into Class I (<50 cm), Class II (50–139.9 cm), Class III (140–179.9 cm), and Class IV (≥180 cm). Animals that could not be sized were designated as “eyes only” (EO). The monitoring was conducted for 3 days in the second week of every January in the years reported in Table 1. The tasks were carried out by one person responsible for the actual counting, another person recording the information, and the boat driver. Using these data, the relative abundance index was calculated in Ind/km [20,21].

Ranching: To identify and collect nests, rounds were made along the edge of the lagoon and both sides of the channel, mounted on horseback, allowing for a higher vantage point to conduct a thorough search, visualizing the maximum number of nests possible. All observed nests were collected by the gauchos, who had been trained for this activity. Subsequently, the eggs were transported to the experimental station for incubation under controlled conditions. The number of identified nests is reported to demonstrate the effects of ranching program implementation and *C. latirostris* head-started releasing on population size.

Data Analysis: Shapiro–Wilk and Levene tests were used to determine normality and homogeneity of variance of the parameters measured (nests, number of caimans). After that, an ANOVA followed by a Tukey’s test was used to detect differences in the number of individuals by size class. These analyses were performed using SPSS software v25 (IBM Corporation, Armonk, NY, USA), and statistical significance was set to α < 0.05.

## 3. Results

Population Surveys: The results of the 33 spotlight surveys (presented as a mean of three surveys per year during 11 non-consecutive years) are shown in Figure 1. In the most recent survey (2018), a total of 243 broad-snouted caimans were observed along the 4.5 km survey route, averaging 54 individuals per kilometer. This represents a notable increase of 42.39% in the population compared to the 2016 survey, which recorded a total of 140 broad-snouted caiman.

Population Size-Class Structure: The maximum number of broad-snouted caiman sighted in each size class was as follows: 185 individuals classified as EO and 52 as Class II, both in the 2018 survey; 33 individuals classified as Class III in the 1997 survey; and 22 as Class IV in the 2001 survey (Table 1). Overall, the size-class structure of the El Fisco population is composed of 50.70% individuals classified as EO, 24.59% as Class II, 16.51% as Class III, and 8.20% as Class IV. The one-way ANOVA test indicated variation in size-class distribution (F = 3.826, *p* = 0.03311, *n* = 33). The Tukey’s test indicated that Class IV is less abundant than Class II (*p* = 0.02531, *n* = 33) (Table 1). Individuals classified as EO were excluded from this statistical analysis. Class I is not typically considered due to the high mortality rate associated with this class [23]. Eyes-only individuals were excluded from the analysis because they could belong to any other size class. On the other hand, when considering Classes III and IV as adults, it can be observed that in the early seasons (1992 to 2005), there is a higher percentage of adult individuals (Class III and IV) than juveniles (Class II). Subsequently, there is an inversion of values, with a higher percentage of juvenile individuals (Class II) than adults (Class III and IV). The entire reserve was thoroughly surveyed for nests, and the absence of Class I specimens detected during surveys is a robust indicator that all nests were harvested.

Number of Nests and Population Size: During the 1996 surveys, the presence of active nests confirmed the reproductive activity of the *C. latirostris* population in El Fisco. The survival of these individuals was validated through an increase in *C. latirostris* sightings in subsequent surveys. This led to an increase in the number of harvested nests to over 100 in the following years. During the period between 2009 and 2011, the quantity of harvested nests did not exceed 70, a phenomenon attributed to the influence of climatic variables on species reproduction. Fluctuations in climatic conditions closely correlated with the viability of nests, also impacting their availability for harvest due to floods or droughts that restricted our access. After unfavorable weather conditions that led to a decrease in the number of nests, the next reproductive season showed a higher nest count. This suggests that a possible compensatory mechanism may be occurring. As seen in the number of nests collected in the 2009 season, it was a result of the beginning of a period of adverse climatic conditions for caiman reproduction. In subsequent years, the climatic adversities were even more intense, leading to the decision not to intervene in the area to avoid further stress on the caiman populations due to the severe drought in the region; this is evidenced by the lack of egg harvest information in the 2010 and 2011 seasons (Table 2, Figure 4).

The number of nests was regulated to no more than 70 for the period between 2012 and 2017. All these efforts have culminated in a significant growth of the *C. latirostris* population in El Fisco, reaching over 240 individuals (Table 1, Figure 4).

## 4. Discussion

Nowadays, there are some concepts on the active management of the different types of programs that pursue some change in the status of the resource, which depend on the general objectives, such as recovering the population situation, controlling its abundance, or making rational use of the species.

Obtaining data on the status of crocodilian populations in managed sites is essential for assessing the sustainability of these programs. However, gathering this information using standard night count monitoring techniques proves to be challenging for species with cryptic behavior like *C. latirostris* [4]. The broad-snouted caiman is recognized for its preference to inhabit densely vegetated and shallow aquatic environments [3]. Additionally, it is rarely observed during daylight hours. This characteristic complicates the gathering of accurate population information, as these areas are frequently nearly inaccessible for transportation mechanisms and people. The night survey method itself has several disadvantages, with a major drawback being the bias in the visual detection of individuals due to both physical and environmental variables affecting the counts [24,25,26].

Despite the potential errors associated with the night count method, another indicator of population status is the long-term data on nest quantities, which allow for the assessment of population trends [27]. Our results show an increase in the number of harvested nests over the years in El Fisco, corroborating the data obtained through night counts. This method is currently used in the USA in the *Alligator mississippiensis* ranching program [27]. Platt et al. [28] assert that annual nest counts seem to be the most suitable method for long-term population monitoring of *Crocodylus siamensis* in Laos, although at least five years of data must be accumulated before statistically detecting population trends. However, it is possible that long-term studies are substantially more important as they provide more robust and realistic information about the state of populations. Furthermore, long-term studies help to identify population trends, detect population declines or increases, and evaluate the effectiveness of conservation measures. By monitoring populations over several years, we can assess the impact of management actions and adapt conservation strategies to ensure the long-term viability of populations.

The progressive rise in the number of identified nests over time, coupled with the observation of animals across various size classes, serves as compelling evidence of population recovery. While there may be more advanced methods to assess population dynamics in El Fisco, the substantial quantity and repetitiveness of information gleaned from night counts and nest surveys have demonstrated their sufficiency under these conditions. These methodologies prove effective in evaluating the population growth over time at this site. Coinciding with findings reported in previous studies by Larriera et al. [22], the results from nocturnal monitoring provide crucial information about the observed animals and enable inferences about the current population structure at the time of the study.

In the beginning, the ranching program was performed only to increase the population, and later on, the sustainable use profile was added. There has been fundamental evidence that it has had a non-detrimental impact on the wild populations; on the contrary, it has been beneficial to the conservation of the species and its habitats.

Under natural conditions, it is estimated that only 30% to 50% of eggs laid during the breeding season successfully hatch at the end of the incubation period [6]. Consequently, it is projected that approximately 10% of the hatched animals manage to survive to complete their first year of life [8]. Although parental care behavior includes nest defense and protection of offspring after hatching [29], significant losses occur during embryonic and perinatal stages due to the presence of predators and often because of environmental or anthropogenic factors (floods, droughts, low temperatures, fires, etc.) [30]. Typically, crocodilians exhibit large clutch sizes; however, this is inversely correlated with the survival rate of offspring. Therefore, specific management strategies can be employed to take advantage of this condition for population enhancement or commercial utilization. The implementation of this mechanism does not generate any negative effects on the species [31] or the environment, but on the contrary, it provides significant opportunities for local communities’ livelihoods. The ranching at any stage of the program did not affect the quota of egg harvesting, as happens in many other similar operations, because the federal and state governments identified that the goal of this management was to help prevent the natural loss of caimans in the wild.

The efficacy of reintroducing juvenile crocodilians into the natural population through ranching programs has been a subject of extensive examination. Detractors posit that specimens bred in captivity and subsequently released into the wild may encounter challenges in terms of survival, potentially stemming from the complexities associated with acquiring hunting skills and adjusting to a novel environment. Nevertheless, Proyecto Yacaré has not only verified the survival but also the reproductive success of individuals released into the wild. The findings of this study demonstrate that the combination of egg harvest to mitigate embryonic losses due to the aforementioned causes, the reintroduction of head-started caimans, and the engagement of local communities through incentive-based conservation initiatives are effective and valuable tools for the recovery and sustainable utilization of wild populations of broad-snouted caiman.

The economic incentive provides vital income to remote communities, motivating them to conserve caimans and nesting habitats. In some areas like the Northern Territory in Australia [32,33] or Cispata Bay in Colombia [34], rural communities have established facilities to carry out the entire process of early ranching, from incubating eggs to raising the stock to a yearling, locally within their communities, and the positive effects on wild crocodiles populations were similar to those evidenced in this case. In addition, the sustainable use of caiman products enables the long-term, incentive-driven preservation of the species, increases local communities’ knowledge about this species, and also increases tolerance towards them, making coexistence easier.

Similarly, robust populations of caimans not only provide significant financial benefits but also create very important cultural and social bonds with communities and a commitment that is reflected in concrete conservation actions, protecting the species and its habitat, which undoubtedly have an impact on the overall conservation of the ecosystem. Moreover, the implementation of conservation and sustainable use initiatives in Argentina has significantly influenced and catalyzed progress in crocodilian research, encompassing an understanding of the ecological system and caiman biology.

## 5. Conclusions

Over a span of more than 25 years, our observations in an experimental site with very low anthropogenic activities revealed a clear positive impact attributed to the implementation of the sustainable use program. This impact is evidenced by a noticeable increase in population growth as observed in both annual egg harvest and night survey data. By extrapolating from these results, it becomes evident that comparable effects may be observed in other sites where the program is implemented.

## Figures and Tables

**Figure 1 animals-14-01288-f001:**
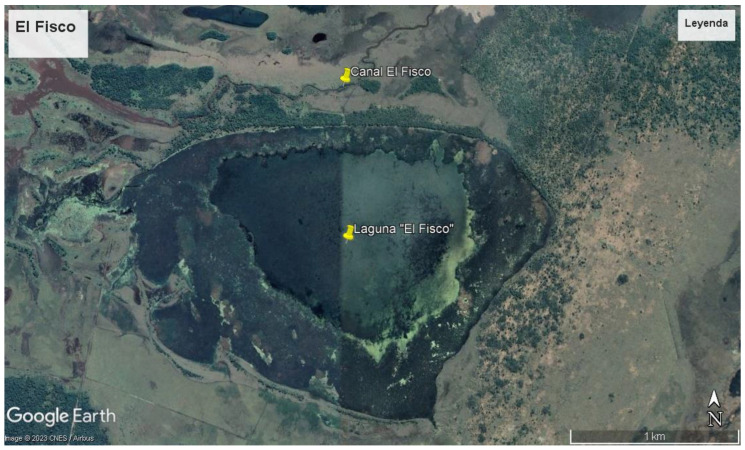
El Fisco Lagoon (in the middle) and Channel (at the top (north)).

**Figure 2 animals-14-01288-f002:**
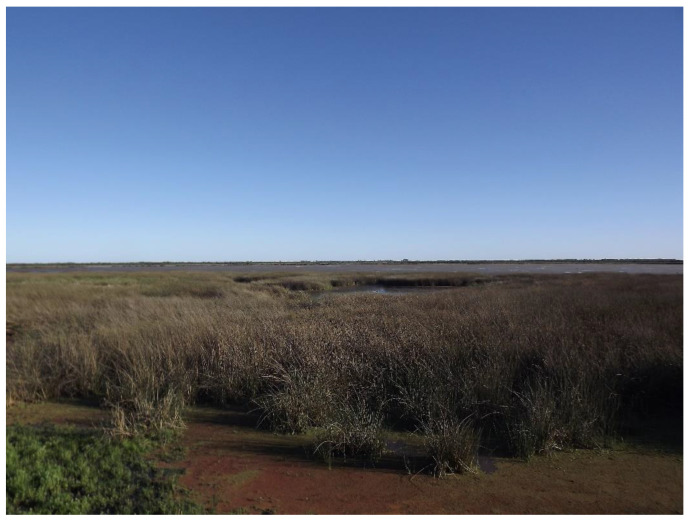
El Fisco Lagoon and the surrounding vegetation area. Photo: H. Ciocan.

**Figure 3 animals-14-01288-f003:**
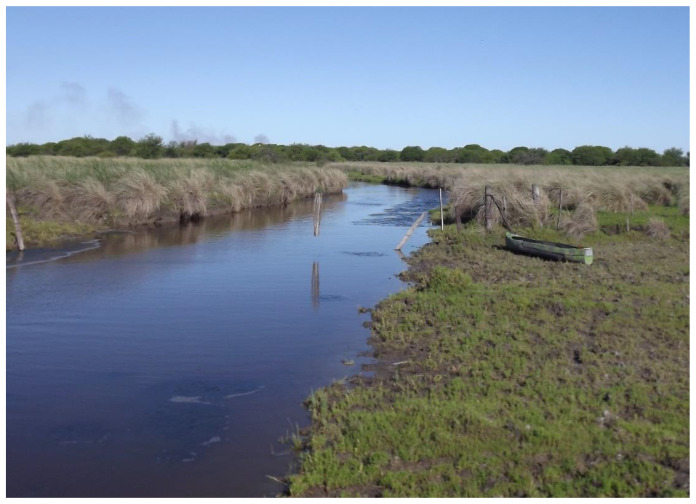
El Fisco Channel. Photo: H. Ciocan.

**Figure 4 animals-14-01288-f004:**
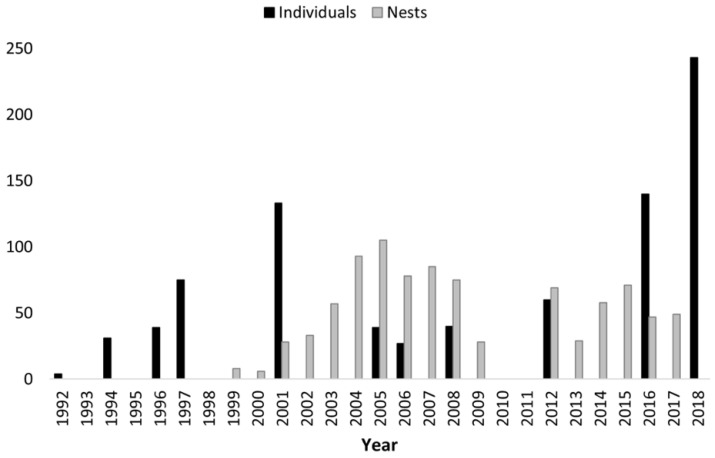
Number of collected nests and caimans sighted through the years.

**Table 1 animals-14-01288-t001:** Population surveys and size-class structure of *Caiman latirostris* in El Fisco Natural Reserve, Santa Fe, Argentina. L: Lagoon; C: Channel. ^a^: Data taken from [22].

Date	No. of Caimans					Distance Surveyed (km)	Caiman Density (Individuals/km)	Location
	Size Class				Total			
	II	III	IV	EO				
1992 ^a^	2	1	1	0	4	12.3	0.325	L, C
1994 ^a^	12	11	3	5	31	12.3	2.52	L, C
1996 ^a^	16	18	2	3	39	12	3.25	L, C
1997 ^a^	14	33	18	10	75	12.3	6.09	L, C
2001 ^a^	14	31	22	76	133	12.3	10.81	L, C
2005 ^a^	9	14	9	12	39	12.4	3.14	L, C
2006 ^a^	18	9	0	0	27	6.7	4.03	L
2008	14	8	5	13	40	4.5	8.89	L
2012	32	7	4	17	60	4.5	13.33	L
2016	27	6	5	112	140	4.5	31.11	L
2018	52	3	4	185	243	4.5	54	L

**Table 2 animals-14-01288-t002:** Number of *Caiman latirostris* harvested nests by local communities in El Fisco per year.

Year	1999	2000	2001	2002	2003	2004	2005	2006	2007	2008	2009	2010	2011	2012	2013	2014	2015	2016	2017	2018
Number of harvested nests	8	6	28	33	57	93	105	78	85	75	28			69	29	58	71	47	49	78

## Data Availability

The data that supported the findings of this study are available in English or Spanish from P.S. upon reasonable request.

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
