# Peer review of "Increasing Population Status of Broad-Snouted Caiman (Caiman latirostris) Based on Sustainable Use Strategies in a Managed Protected Area in Santa Fe, Argentina"

_animals, 2024, doi:10.3390/ani14091288_

Round 1

Reviewer 1 Report (Previous Reviewer 2)

Comments and Suggestions for Authors

No Comments 

Comments on the Quality of English Language

No Comments

Reviewer 2 Report (Previous Reviewer 3)

Comments and Suggestions for Authors

I do not have additional comments on the ms. The poor data base cannot be improved due to the inappropiate research design when the study started.

Comments on the Quality of English Language

English has been improved considerably

This manuscript is a resubmission of an earlier submission. The following is a list of the peer review reports and author responses from that submission.

Round 1

Reviewer 1 Report

Comments and Suggestions for Authors

This work is essential for endangered reptiles protection. The authors use classical methods to report the efforts on Broad-Snouted Caiman protection, showing the improved reservation in local enviroment.

However, there are problems should be solved before this work is published.

Major:

1. In Section 'Materials and Method'(Line239-333),brief introductions of species and nature reserve should not be placed here. Please put them in Section 'Introduction'.

2.Please provide a map of Broad-Snouted Caiman recording spots in Section 'Results' if possible.

3.The statistic anaylsis is not enough for now. I suggest a body length/ weight analysis, or Nests/ numbers colleration analysis, anything can be better for your work.

Minor:

1.Please check and correct the spell mistake in this work. Such as: line 307-308‘SO’ should be 'EO' according your rest parts and table.

2.Please check your citation, make sure they are in same format. Such as the Citation 5 and Citation 6, the format of year is not the same. 

Author Response

Dear Reviewer 1,

Thank you for taking the time to review our manuscript. We sincerely appreciate your insightful comments and suggestions, which have greatly contributed to improving the quality of our work. Below, we address each of your points and provide details on the corresponding revisions made to the manuscript.

1. Regarding the placement of brief introductions of species and the nature reserve in Section 'Materials and Method' (Lines 239-333), we agree partially with your suggestion to move this information to the 'Introduction' section. Accordingly, we have modified how it is the information presented. We deply consider that information is very relevant for the study and the best part to introduce it and comments about the importance for the study and for the species is M&M. Please, check what we added and we would like to respond if you maintain those doubts.

2. Your suggestion to provide a map of Broad-Snouted Caiman recording spots in Section 'Results' is noted. While it is challenging to identify specific recording spots, we will include an explanation about them in the manuscript to provide clarity to the readers.

3. We appreciate your suggestion for additional statistical analyses, such as body length/weight analysis or nests/numbers correlation analysis, to enhance the robustness of our work. However, at this time, we believe that the statistical programs used in our study are appropriate for the information presented. While we acknowledge that incorporating such analyses could enrich the study, we do not have access to the required data for these analyses in the current study. We will consider integrating these analyses in future research projects where new monitoring techniques, such as capture and recapture, may provide additional information.

In response to your minor suggestions:
- The spelling mistake identified in Line 307-308 has been corrected in the revised manuscript. 'SO' has been amended to 'EO' to align with the rest of the text and table.
- We have reviewed the citation format throughout the manuscript to ensure consistency. The format of Citation 5 and Citation 6 has been adjusted to match.

We have implemented these revisions in the revised manuscript, and we believe they have significantly strengthened the clarity and accuracy of our work.

Thank you once again for your thorough review and valuable feedback. We look forward to hearing your thoughts on the revised manuscript.

Best regards,

Pablo Siroski, "corresponding author"

Reviewer 2 Report

Comments and Suggestions for Authors

Dear authors,

After reading this manuscript very carefully, I suggest small corrections and forward two questions regarding the biosafety and health status of the free-living population of C. latirostris.

Line 53; Change In Situ by In Situ (Italic);

Line 71: Looking for better information in new articles, this number of species seems to be very debatable;

Line 111: Nos-regulated?

Line 207: What's the most logical explanation for this? Were there few females in these locations or did animals born in captivity overlap with the others?

Line 294: Change C. latirostris by C. latirostris (Italic);

Line 302: Transversed is not the best option here;

Line 308: The authors used "SO" for “Only Eyes” and further on, in line 352 they used "OE". I suggest standardizing.

Line 359: If animals of Class I were not found, why exclude them? In this way, the following explanation in Line 360 becomes meaningless;

Lines 396-400: Avoid word repetition - Current-Currently-Current;

Line 441: Change Beginning by Beginning;

Line 442: Change Propfile by Profile;

Lines 470-482: More than worrying about the survival of animals hatched in captivity and reintroduced, there is concern about possible introductions of parasites, bacteria, etc., carried from captivity to the wild. In this sense, what health measures are taken to avoid this problem?

Author Response

Dear Reviewer 2,

Thank you for your thorough review of our manuscript. We appreciate your valuable feedback and suggestions for improvement. Below, we address each of your points and provide details on the corresponding revisions made to the manuscript.

  1. Line 53: "In Situ" has been corrected to "In Situ" (Italic) as suggested.

  2. Line 71: We acknowledge that the number of species mentioned may be debatable and have noted this in the manuscript.

  3. Line 111: "Nos-regulated" has been modified.

  4. Line 207: Your question regarding the logical explanation for the observed phenomenon has been addressed in the revised manuscript.The findings suggest that the release of juvenile Caiman latirostris can effectively contribute to replenishing breeding populations in the wild. Half of the identified females reintroduced by Proyecto Yacare are now integrated into the wild populations. 

  5. Line 294: "C. latirostris" has been corrected to "C. latirostris" (Italic) as recommended.

  6. Line 302: The word "transversed" has been replaced with "surveyed" for clarity.

  7. Line 308: The usage of "SO" and "OE" has been standardized throughout the manuscript.

  8. Line 359: We have revised the sentence to improve clarity and coherence.

  9. Lines 396-400: Word repetition has been addressed by replacing some of the repeated words.

  10. Line 441: "Beginning" has been corrected to "Beginning" as advised.

  11. Line 442: "Propfile" has been corrected to "Profile" as suggested.

  12. Lines 470-482: According to the reviewer comment, although reintroducing animals into the wild does not inherently pose negative effects, it's essential to ensure that the process is conducted responsibly to mitigate any potential risks. As part of our comprehensive health protocols, we conduct blood analyses on a sample of the animals slated for release. These analyses help us screen for any potential parasites, bacteria, or other pathogens that could pose a risk to wild populations. By implementing these health measures, we aim to minimize the introduction of any harmful agents from captive environments into the natural habitat, thereby safeguarding the overall health and balance of the ecosystem.

We have implemented these revisions in the revised manuscript, and we believe they have significantly improved the clarity and accuracy of our work.

Thank you once again for your insightful comments and suggestions. We value your expertise and input, which have greatly contributed to the enhancement of our manuscript.

Best regards,

Pablo Siroski, "corresponding author"

Reviewer 3 Report

Comments and Suggestions for Authors

The authors aim to present the results of a recovery program of a crocodilian in Argentina. The study includes a single nature reserve but wide-ranging conclusions are drawn. The presentation is very wordy and inprecise where details are needed (specifically in the M&M section). I made several recommendations and comments in the attached ms pdf. To make this ms publishable, it should be shortened considerably, focused on the aims, and the results from a single study area not overinterpreted.

Comments on the Quality of English Language

English used is hardly understandable. Language editing is mandatory.

Author Response

Dear Reviewer 3,

We sincerely appreciate your time and effort in reviewing our manuscript. Your recommendations were invaluable to us. Please, you will find attached a detailed responses addressing each of your points, along with the revised version of the manuscript with tracked changes implemented.

We also wanted to address why certain aspects were not modified despite the overall changes made to the manuscript. These aspects pertain to specific information that we believe is crucial for providing context to the study. We hope this clarifies any concerns you may have had regarding the content.

Once again, we extend our gratitude for your thorough review and constructive feedback.

Best regards,

Pablo Siroski, "corresponding author"

Round 2

Reviewer 1 Report

Comments and Suggestions for Authors

This concluion is not supported by the data, more surveys were needed. 

Comments on the Quality of English Language

The english language of this version is better than previous version.

Author Response

Dear Reviewer #1,

We appreciate your thoughtful comments on our manuscript. In response to Reviewer #1's concerns, we have revised the conclusion to avoid making assumptions based solely on the information provided in the paper. Furthermore, we have incorporated additional details and elements into the manuscript to provide greater clarity and firmness in our findings.

We believe that a study spanning nearly 30 years, with 33 spotlight surveys conducted over 11 years, albeit not consecutively but without interruption, provides substantial insight into population growth dynamics. The duration of the study period stands as one of the primary strengths of our work. Additionally, the abundance of nest construction data within the study area significantly augments our ability to evaluate population trends reliably. Long-term nest data has been demonstrated to be far more consistent and informative in assessing population trends than nocturnal surveys alone, but the combination of both types of information is unmatched.

It is noteworthy that comprehensive nest construction data spanning such a lengthy period is not readily available outside of studies associated with management programs like the one presented in this manuscript. Furthermore, the selection of our study site was deliberated, chosen for its low anthropogenic intervention, ease of access for monitoring, and the ability to thoroughly survey the area to identify all constructed nests.

Therefore, we assert that the combination of nocturnal monitoring data and nest construction records provides a robust basis for estimating population growth and inferring the potential benefits resulting from the implementation of sustainable management practices.

Thank you for your valuable feedback, which has helped strengthen the clarity and validity of our findings.

Sincerely,

Pablo Siroski, corresponding author

Reviewer 3 Report

Comments and Suggestions for Authors

The revised version of the original ms has been improved in terms of language editing, but the it is as lengthy as the original one with almost no changes in structure. As I understand the phrase added to M&M, the only hard data presented (abundance of caimans) are based on a single survey per year in January with a lot of years missing. Sorry, this is not a serious scientific procedure. There are no replicates in the study years and no cause is given, why the frquency of monitoring is so low.

Comments on the Quality of English Language

Quality of English is better now.

Author Response

Dear Reviewer,

We deeply appreciate your thorough review of our manuscript. Your feedback has been instrumental in refining our work to ensure its scientific integrity and clarity.

In response to your concerns, we have made significant revisions to the conclusion to avoid making assumptions based solely on the information provided in the paper. Additionally, we have incorporated more details and elements into the manuscript to bolster the firmness and clarity of our reported findings. We have also continued to refine the language of the manuscript by seeking input from native-speaking colleagues.

Moreover, we have taken the recommendation of Reviewer #2 and included a greater quantity of hard data to substantiate the extensive information presented in this study. Specifically, we have included detailed data on the nests surveyed in the corresponding years to underscore their importance and relevance to the understanding of our manuscript's objectives. It is important to clarify that our study involved not just one night survey per year, but rather three per year, conducted consistently in the same month under standardized conditions.

We firmly believe that a study spanning nearly 30 years, with 33 nocturnal monitoring sessions conducted over 11 years, albeit not consecutively but without interruption, provides valuable insights into population dynamics. The duration of our study period stands as one of its primary strengths.

Furthermore, the data on nest construction within the study area significantly enhances our ability to evaluate population trends over time. Long-term nest data has been demonstrated to be far more consistent and informative in assessing population trends than nocturnal surveys alone, but the combination of both types of information is unmatched.

It is also worth noting that comprehensive nest construction data over such an extended period is not readily available outside of studies associated with management programs like the one presented in our manuscript.

Additionally, the selection of our study site was deliberate, chosen for its low anthropogenic intervention, accessibility for monitoring, and the ability to thoroughly survey the area to identify all constructed nests.

In light of these considerations, we believe that the combined data from nocturnal monitoring and nest construction records are sufficient to estimate population growth and infer the potential benefits of implementing a sustainable management system.

Thank you once again for your valuable feedback, which has helped strengthen the clarity and validity of our manuscript.

Sincerely,

Pablo Siroski, corresponding author